# Enhancing Plant Resistance to Sri Lankan Cassava Mosaic Virus Using Salicylic Acid

**DOI:** 10.3390/metabo15040261

**Published:** 2025-04-10

**Authors:** Chonnipa Pattanavongsawat, Srihunsa Malichan, Nattachai Vannatim, Somruthai Chaowongdee, Nuannapa Hemniam, Atchara Paemanee, Wanwisa Siriwan

**Affiliations:** 1Department of Plant Pathology, Faculty of Agriculture, Kasetsart University, Bangkok 10900, Thailand; chonnipa.pa@ku.th (C.P.); srihunsa.m@ku.th (S.M.); nattachai.va@ku.th (N.V.); somruthai.ch@ku.th (S.C.); nuannapa_joy@hotmail.com (N.H.); 2National Omics Center (NOC), National Center for Genetic Engineering and Biotechnology (BIOTEC), National Science and Technology Development Agency (NSTDA), Pathum Thani 12120, Thailand; atchara.pae@biotec.or.th; 3Food Biotechnology Research Team, Functional Ingredients and Food Innovation Research Group, National Center for Genetic Engineering and Biotechnology (BIOTEC), Thailand Science Park, Pathum Thani 12120, Thailand

**Keywords:** salicylic acid, cassava mosaic disease, Sri Lankan cassava mosaic virus, plant immune response, induced resistance

## Abstract

**Background:** Cassava mosaic disease (CMD), caused by the Sri Lankan cassava mosaic virus (SLCMV), significantly increases cassava yield losses in Thailand, with losses ranging from 30% to 80%, and is exacerbated by limited access to healthy planting materials. **Methods:** This study explored salicylic acid (SA) as a potential treatment for enhancing disease resistance in CMD infected cassava plants. SA was applied at 100 and 200 mg/mL, and its effects were evaluated using quantitative real-time polymerase chain reaction (qPCR) and reverse transcription qPCR (RT-qPCR) to measure viral loads and the expression levels of resistance genes. **Results:** Although SA treatment did not considerably affect disease severity, foliar CMD symptoms visibly decreased, particularly with 200 mg/mL SA, which also reduced SLCMV particle counts at 1- and 2-weeks post-treatment. SA upregulated the expression of pathogenesis-related proteins (PRs), including *HSP90.9*, *WRKY59*, *SRS1*, and *PR9e*. Additionally, SA enhanced the regulation of secondary metabolite pathways involving L-serine within the glycine, serine, and threonine metabolism, as well as the phenylpropanoid biosynthesis pathways. **Conclusions:** These findings collectively indicate that SA enhances resistance through the systemic acquired resistance (SAR) pathway and can serve as a potential strategy for the management of CMD, particularly in regions where healthy cassava planting materials are scarce. The study highlights the efficacy of SA in reducing viral particles, inducing the immune response, and providing a promising approach for controlling CMD.

## 1. Introduction

Cassava (*Manihot esculenta* Crantz) is a crucial root vegetable worldwide, known for its high carbohydrate content and role as a primary energy source [1]. In Thailand, cassava holds a major position in the agricultural sector, and the area of cassava plantation in Thailand is the second largest worldwide, after Nigeria. Cassava production in Thailand exceeds 31 million tons, which contributes considerably to the economy, and the annual yield and export value of cassava production exceeded 4.27 million tons and 2427 million dollars, respectively, in 2023. Despite its economic importance, the commercial varieties of cassava are susceptible to various diseases that can significantly reduce the yield. The cassava plantations in Thailand are vulnerable to various diseases, including cassava bacterial blight, brown leaf spot, blight leaf spot, stem rot, cassava mosaic disease (CMD), and witch’s broom disease [2], of which CMD poses a severe threat, substantially reducing cassava production by 30–80% [3].

CMD is caused by the cassava mosaic virus, which belongs to the *Begomovirus* genus under the *Geminiviridae* family [4]. To date, 11 species of CMV have been reported globally, with Sri Lankan cassava mosaic virus (SLCMV) being the predominant species in Thailand [5]. The first outbreak of CMD was identified in Thailand in August 2018 in a region bordering Cambodia [6], where previous CMD outbreaks had been reported [7]. CMD is transmitted by the tobacco whitefly (*Bemisia tabaci*) as well as through infected stem cuttings. Whitefly can efficiently spread the virus within crop field [8]. After feeding on CMD -infected plants for 4–6 h, the insect becomes capable of transmitting the virus. Although it is possible that the virus replicates within the insect vector, the exact mechanism remains unclear. Transmission to a healthy plant can occur within as little 15 min of feeding, with longer feeding duration increasing the likelihood of infection, thereby facilitating rapid and widespread disease dissemination [9,10]. Propagation through stem cutting is the common method used in cassava cultivation [11]. The *Geminiviridae* viruses present in the phloem of CMD- infected cuttings can contribute to the spread of the disease across fields and over long distances [12]. Typical symptoms include light green or yellow and dark green mosaic patterns on young shoots and new growth. Infected plants exhibit stunted trunks and growth retardation, leading to a reduction in cassava root production by over 70% [13].

Plants respond to diseases via various mechanisms such as induced resistance (IR), which is mediated by structural and biochemical defenses, including the synthesis of a wax or cutin coat and phytoalexins. Pathogenesis-related proteins (PRs) play a key role in the process of IR by retarding infections without harming the plant. Plants can also enhance disease resistance using biotic elicitors derived from pathogens [14,15]. Systemic acquired resistance (SAR) is another crucial mechanism of resistance in which plants respond to elicitors by producing proteins and chemicals that inhibit the growth of pathogens to confer sustained resistance [16]. Phytohormones are key signaling substances in plant resistance, of which salicylic acid (SA) plays a crucial role in inducing resistance to pathogens and signaling [17,18].

SA is a naturally occurring phytohormone that regulates various aspects of plant physiology. It influences plant growth, stomatal activity, nutrient absorption, and the synthesis of chlorophyll, phenolic compounds, amino acids, and antioxidants [19]. SA also acts as a signaling molecule, inducing the expression of genes that enhance resistance to pathogens [20]. In plants, SA is synthesized from the amino acid phenylalanine, primarily by the action of phenylalanine ammonia lyase and isochorismate pyruvate lyase [21]. During infection, SA production increases, leading to the activation of non-expressor of pathogen-related gene 1 (*NPR1*). *NPR1,* in turn, triggers the accumulation of several PRs, including *PR-1*, *PR-2*, *PR-4*, *PR-5*, *PR-8*, and *PR-11*. This cascade induces SAR and enhances resistance against various pathogens, including tobacco mosaic virus (TMV) and the fungus, *Colletotrichum lagenarium* [22,23,24].

The study by Király et al. [25] demonstrated that the application of sufficient quantities of sulfate to tobacco plants reduced the viral loads and increased the expression of *PR1* and *PR2*, indicating an enhanced resistance to TMV. The application of sulfate induced the production of SA and triggered the defense mechanisms of tobacco plants. Wang et al. [26] investigated the effect of spraying SA, jasmonic acid (JA), and coronatine (COR) on potato plants infected with tomato yellow leaf curl virus (TYLCV). The findings revealed that treatment with 0.5 mM SA, 0.5 mM JA, and 0.1 µM COR significantly reduced the viral load compared to the control. The reduction was most significant following treatment with JA, followed by SA and COR. These compounds also upregulated the genes that were involved in pathogen response and reduced the elevated viral load.

Recent studies on plant metabolism provide insights into the differential expression of plant traits, including physiology, yield, and photosynthesis, across different cultivars. For instance, the study by Obata et al. [27] analyzed the metabolites in six commercial cultivars of cassava from Africa to investigate the cultivar-specific factors contributing to high yields. The findings indicated that traits such as carbon assimilation rate, K battery, root starch synthesis, and the accumulation of trehalose and chlorogenic acid were strongly associated with high yield. Chaowongdee et al. [28] additionally analyzed the metabolite profiles of different cultivars of SLCMV-infected and healthy cassava plants. The results demonstrated that flavonoid biosynthesis, as well as the phenylalanine, tyrosine, and tryptophan biosynthesis pathways, contribute to SAR.

The present study aimed to determine the concentration at which SA can induce resistance to CMD in infected cassava plants. Resistance to CMD can decrease the viral load and induce the expression of genes related to SAR and defense mechanisms. These findings can provide novel insights into the role of SA in inducing resistance in cassava plants and facilitating its application for the management of CMD.

## 2. Materials and Methods

### 2.1. Plant Materials, SA Treatment, and Collection of Leaf Samples

The KU50 landrace cultivar of cassava infected with CMD was vegetatively propagated. The stems were cut into sections with 3–4 buds per segment and planted in a greenhouse at the Department of Plant Pathology, Faculty of Agriculture, Kasetsart University, Thailand. The temperature of the greenhouse was maintained at 27–29 °C, under 14 h of light, and a relative humidity of 70–80%. After six weeks of planting, the leaves were sprayed with SA (Acros Organics, Shanghai, China) dissolved in 60% ethanol, and then diluted with sterile water to final concentrations of 100 and 200 mg/mL. The control plants were treated with 60% ethanol. Each treatment was replicated using five plants, and each plant was sprayed with 1 mL of SA solution. Leaf samples were collected at 0, 1, 2, 3, and 4 weeks post-treatment, and stored at −20 °C. The leaf samples were used to measure the viral particles counts and expression of resistant genes through quantitative real-time polymerase chain reaction (qPCR) and reverse transcription qPCR (RT-qPCR) analyses.

### 2.2. Disease Severity

The symptoms of CMD were assessed from the photographs captured at 0, 1, 2, 3, and 4 weeks post-treatment. The disease severity was evaluated according to the method described by Sseruwagi et al. [29], in which the symptoms are classified into five levels: 1 = no symptoms, 2 = mild chlorotic patterns, 3 = mosaic patterns on all the leaves and foliar distortion, 4 = mosaic pattern on all the leaves, foliar distortion, and a general reduction in leaf size, and 5 = misshapen and twisted leaves and stunting of the entire plant. The disease-severity percentage was calculated using the following formula by Olasanmi et al. [30]:$$\text{Disease severity}=\frac{\text{Sum of all disease ratings}}{\text{Total number of plants}\times \text{maximum disease rating}}\times 100$$

### 2.3. DNA Extraction

The samples of cassava leaves were used to extract DNA for quantifying the number of SLCMV particles in each of the treatment groups. DNA extraction was performed using the cetyl trimethylammonium bromide (CTAB) method of Doyle and Doyle (1987) Doyle and Doyle [31], with slight modifications. Cassava leaves (100 mg) were mixed with 700 µL CTAB buffer (1 M Tris-HCl (pH 8), 1.4 M NaCl, 0.5 M ethylenediaminetetraacetic acid (EDTA), 4.0 g CTAB, and 2% polyvinylpyrrolidone (PVP)) and crushed. The mixture was then incubated at 65 °C for 30 min, following which 700 µL chloroform/isoamyl (24:1) was added to the tubes. The mixture was then centrifuged at 12,000 rpm for 10 min, following which 400 µL of the supernatant was transferred to a clean 1.5 mL tube. The supernatant was mixed with 400 µL isopropanol and centrifuged at 12,000 rpm for 10 min. The supernatant was discarded, and the DNA pellet was washed twice with 400 µL of 70% ethanol, dried at room temperature for 1–2 h, and then resuspended in 30 µL of nuclease-free water (Thermo Fisher Scientific, Waltham, MA, USA). The extracted DNA was visualized by 1.5% agarose TAE gel electrophoresis (100 V for 30 min) using RedSafe Nucleic Acid Staining Solution (iNtRON Biotechnology, Sangdaewon, South Korea) in 0.5× TAE buffer (1 M Tris-HCl (pH 8), 0.5 M EDTA, and glacial acetic acid). The images of the gels were analyzed using the Syngene software v1.8.13. (Synoptics Ltd., Cambridge, UK). The DNA solutions were stored at −20 °C for further experiments.

### 2.4. Quantification of SLCMV Particles

Amplification and quantification were performed using the SYBR Green dye for measuring the viral particle using qPCR, which was performed on a CFX96 Connect RT-PCR System (Bio-Rad, Hercules, CA, USA). For qPCR, 0.5 µL of the 5′-TAAGAGGTTTTGCGTTAAGTCCG-3′ and 5′-TGTACAGCATCAATGCATTCTCG-3′ forward and reverse primers specific for the SLCMV coat protein, which is encoded by the *AV1* gene, were used along with 3 µL of nuclease-free water, 1 µL of 100 ng/mL DNA template, and 5 µL of 2x qPCRBIO SyGreen Mix Lo-ROX (Copenhagen Biotech Supply, Bronshoj, Denmark). The conditions of qPCR amplification included pre-incubation at 95 °C for 2 min, followed by 40 cycles of denaturation at 95 °C for 5 s, annealing at 60 °C for 30 s, and extension at 65 °C for 5 s. Fluorescence was measured at the end of each cycle, and the melting curve was analyzed between 65 °C and 95 °C with a 10 s hold at increments of 1 °C. Each reaction was performed in triplicate, and the standard curve was prepared for each experiment by co-amplifying known quantities of the SLCMV coat protein. Seven consecutive dilutions (dilution factor 1:10) of the SLCMV coat protein, containing 10^6^ to 10^1^ copies per reaction, were prepared. The quantity of the SLCMV coat protein DNA in the sample was determined by plotting the threshold cycle (Ct) values onto the standard curve.

### 2.5. RNA Extraction and cDNA Synthesis

RNA was extracted from the samples of cassava leaves sprayed with SA and the control solution after 0, 1, 2, 3, and 4 weeks of treatment, according to the method described by Behnam et al. (2019) Behnam et al. [32]. To this end, 100 mg of the leaf samples was crushed to a powder using liquid nitrogen and transferred to a 2 mL tube, following which 700 µL of the pre-heat extraction buffer (100 mM Tris-HCl (pH 8), 100 mM NaCl, 25 mM EDTA, 1% sodium dodecyl sulfate (SDS), 2% PVP, and 2% mercaptoethanol) was added to the powdered sample. The mixture was vortexed and 700 µL of chloroform was added to the sample, followed by centrifugation at 15,000 rpm for 10 min. The supernatant was transferred to a clean 2 mL tube, to which 600 µL chloroform was added, and the mixture was centrifuged at 15,000 rpm for 10 min, and the process was repeated twice. The supernatant was transferred to a clean 1.5 mL tube, to which 0.33 volumes of 8 M lithium chloride were added, and the mixture was allowed to incubate overnight at 4 °C. The mixture was then centrifuged at 15,000 rpm for 10 min, and the supernatant was discarded. The pellet was dissolved in 500 µL diethyl pyrocarbonate (DEPC) and centrifuged at 15,000× *g* for 10 min, following which one volume of phenol: chloroform: isoamyl alcohol (25:24:1) and one volume of chloroform: isoamyl alcohol (24:1) was added to the mixture. The RNA pellet was then precipitated overnight with 500 µL of 5 M NaCl at −80 °C. The precipitated tube was centrifuged at 15,000 rpm for 10 min and the supernatant was discarded. The RNA pellet was washed with 70% ethanol, air-dried on ice for 10 min, and subsequently resuspended in 30 µL of nuclease-free water. The extracted RNA was stored at −80 °C, and its purity was evaluated based on the A260:A280 and A260:A230 absorbance ratios using a NanoDrop^®^ ND-1000 spectrophotometer (Thermo Fisher Scientific, Waltham, MA, USA). The quality of the extracted RNA was assessed by 1.5% agarose TAE gel electrophoresis (100 V for 30 min) using RedSafe Nucleic Acid Staining Solution (iNtRON Biotechnology, Sangdaewon, Republic of Korea) in 0.5x TAE buffer (1 M Tris-HCl (pH 8), 0.5 M EDTA, and glacial acetic acid). A 1 kb DNA ladder (Thermo Fisher Scientific, USA) was used as reference. The images of the gels were subsequently analyzed using the Syngene software v1.8.13. (Synoptics Ltd., Cambridge, UK).

In this study, cDNA was synthesized using 1 µL of RevertAid reverse transcriptase (Thermo Fisher Scientific, Waltham, MA, USA), 1 µL of RiboLock RNase Inhibitor, 2 µL of 10 mM dNTP mix, 1 µL of 10 mM Oligo(dT), 4 µL of 5× reaction buffer, 10 µL water, and 1 µL of 100 ng/µL RNA template. The cDNA products were stored at −20 °C for further experiments.

### 2.6. RT-qPCR

The *PDF2*, *PR1*, *PR7f5*, *PR9e*, *SYP121*, *WRKY89*, *Hsp90.9*, *Hsf8*, and *SRS1* genes were amplified to evaluate the relative expression levels of the genes that were involved in the induction of plant defenses by SA. The *UBQ10* gene served as the reference for the RT-qPCR assays. The nucleotide sequences of the primers are provided in Appendix A. RT-qPCR amplification was performed using a CFX96 Connect RT-PCR System (Bio-Red, Hercules, CA, USA). The 10 µL reaction mixtures comprised 5 µL of 2x qPCRBIO SyGreen Mix Lo-ROX (PCR Biosystems, London, UK), 0.5 µL of each of the forward and reverse primers, 1 µL of 100 ng/µL cDNA template, and 3 µL of nuclease-free water The conditions for thermocycling were as follows: initial denaturation at 95 °C for 2 min, followed by 40 cycles of denaturation at 95 °C for 5 s, annealing at 60 °C for 30 s, and extension at 65 °C for 5 s. The fluorescence signals were recorded at the end of each cycle. The melting curve was analyzed from 65 °C to 95 °C, with a 10 s hold at increments of 1 °C for verifying the specificity of the amplification. Each reaction was performed in triplicate to ensure reproducibility. The relative mRNA abundance of the target genes was normalized to that of *UBQ10*, and the fold change (log_2_) in gene expression was calculated using the 2^−∆∆CT^ method [33] (Rao et al., 2013).

### 2.7. Metabolomic Analysis

The metabolites were extracted according to the method described by Chaowongdee et al. (2023) [28], and UHPLC-HRMS/MS was performed for analyzing the metabolites. A 200 ppm cassava metabolite extract (2 µL) was injected onto a Hypersil GOLD™ Vanquish C18 column (2.1  ×  100 mm, 1.9 µm, Thermo Fisher Scientific, USA) equipped with a guard column, which was maintained at 40 °C and operated at a flow rate of 0.4 mL/min. Mobile phase A consisted of 0.1% FA in water, and mobile phase B consisted of 0.1% FA in acetonitrile. The gradient was initiated with 5% B for 4 min, increased to 90% B over 10 min, held at 90% B for 4 min, returned to 5% B over 1 min, and finally equilibrated for 25 min. Mass spectrometry was performed using a Q-Exactive HF-X Orbitrap system (Thermo Fisher Scientific, USA) equipped with a heated electrospray ionization source, and the positive and negative ions were acquired in the full-scan MS1/data-dependent MS2 (dd-MS2) mode. The following settings were used for mass spectrometry: spray voltage of 3.5 kV (positive) and 2.5 kV (negative); sheath gas, 45 AU; auxiliary gas, 10 AU; sweep gas, 2 AU; capillary temperature, 250 °C; MS1 resolution, 120,000; dd-MS2 resolution, 30,000; scan range, 100–1500 *m*/*z*; AGC target, 3e6; max injection time, 100 ms; and stepped NCE of 20, 30, and 40 eV. The data were processed with Compound Discoverer, whereas compound annotation was performed using the mzCloud, mzVault, and ChemSpider databases.

The metabolite data were analyzed using MetaboAnalyst 6.0 (https://www.metaboanalyst.ca/, accessed on 31 October 2024) The significant differences between the mean values of the different groups were determined using Fisher’s least significant difference (LSD) test, and *p* ≤ 0.05 considered statistically significant [34]. The functions and pathways associated with the differential compounds were identified through Kyoto Encyclopedia of Genes and Genomes (KEGG) pathway analysis (https://www.genome.jp/kegg/, accessed on 31 October 2024) [35].

### 2.8. Statistical Analyses

The data obtained from five biological replicates and three technical replicates were analyzed for viral particle quantification, disease severity, relative gene expression levels, and regression analysis using one-way analysis of variance (ANOVA) in the RStudio program v2024.12.1+563. Significant differences among the means of different groups were determined using the Least Significant Difference (LSD) test, with *p* ≤ 0.05 considered statistically significant.

## 3. Results and Discussion

### 3.1. Effects of SA Against SLCMV

The effects of SA on the percentage of disease severity in CMD-infected plants were evaluated at SA concentrations of 0, 100, and 200 mg/mL. The infected cassava plants were sprayed with different concentrations of SA at 0, 1, 2, 3, and 4 weeks post-treatment. The results of statistical analysis indicated that there were no significant differences in the percentage of disease severity among the different treatment groups across all the time periods tested herein. The results are provided in Figure 1. Interestingly, disease severity decreased after 1 and 2 weeks of spraying SA. As depicted in Figure 2, the reduction in disease severity visually correlated with the absence of the foliar symptoms of CMD. These findings align with the results of the study by Sofy et al. [36], which reported that the disease severity in eggplants infected with *Alfalfa mosaic virus* [37] decreased by 68.34% after 3 weeks of treatment with SA. Similarly, Basit et al. [38] observed that the disease severity in tobacco plants infected with TMV reduced following the application of 60% SA. These results demonstrated that SA effectively reduced the severity of CMD, and that the symptoms improved after 1 and 2 weeks of treatment. This indicated that, although the plants were infected with SLCMV, which had already triggered SAR before treatment, the application of SA further enhanced SAR in infected cassava plants. The application of SA further triggered SAR in the infected plants, which induced a more rapid defense response against SLCMV.

Regarding the effect of SA on endogenous SA content, some studies have indicated that the application of exogenous SA can increase endogenous SA levels, thereby enhancing the plant’s defense mechanisms. However, higher concentrations of SA can sometimes induce a feedback mechanism that reduces endogenous SA production. The impact of exogenous SA may also vary based on the plant’s developmental stage and its ability to regulate hormonal levels. While literature specific to the effect of exogenous SA treatment on endogenous SA in cassava is limited, similar findings have been reported in other crops, such as tomato and tobacco [38]. Further research is needed to better understand the dynamics of SA regulation in cassava during virus-induced stress and hormonal treatment.

The viral count in cassava plants infected with CMD following treatment with SA at concentrations of 0, 100, and 200 mg/mL was measured by qPCR. The results indicated that treatment with 200 mg/mL SA significantly reduced the number of SLCMV particles at 1 and 2 weeks post-treatment compared to that before treatment. In contrast, the viral particle counts gradually decreased at 1, 2, and 3 weeks post-treatment with 100 mg/mL SA. Meanwhile, the number of viral particles in the control group increased steadily during the same period (Figure 3). The trend in the number of SLCMV particles following the application of SA was examined by regression analysis, and the results indicated that the viral counts decreased following treatment with SA, and assumed the equation b = −8.07 × 10^11^ ± 3.12 × 10^11^ after treatment with 100 mg/mL SA and b = −2.68 × 10^11^ ± 1.77 × 10^11^ after treatment with 200 mg/mL SA, where b denotes the slope of viral load versus time (Appendix A). These results suggest that the application of 200 mg/mL SA would be effective for inducing disease resistance in cassava plants infected with CMD, which aligns with the findings of other studies. For instance, Li et al. [39] observed that the viral load in tomato leaves inoculated with TYLCV decreased from days 2 to 10 after spraying SA, but increased on day 14. The study also revealed that SA IR to yellow leaf curl disease in tomatoes after approximately 10 days of treatment. The study by Sofy et al. (2021) [36], additionally revealed that the treatment of tomato plants with SA before inoculating with TYLCV reduced by the viral count in SA-treated plants by 74.6% compared to that of the control on day 7 post-treatment, indicating that SA can effectively reduce the viral load in infected plants. The reduction in disease severity and viral loads could be attributed to the activation of the hypersensitive response in plants by SA, which induces the death of infected tissues to eliminate viruses [40,41]. SA can also inhibit viruses by inducing the generation of reactive oxygen species (ROS), a key defense mechanism in plants. This in turn triggers stomatal closure, induces programmed cell death, and activates various genes to trigger pathogen resistance [42].

These findings suggested that treatment with SA, especially at a concentration of 200 mg/mL, enhances SAR in cassava plants even after infection with SLCMV. The observed reduction in viral load and disease severity highlights the potential of SA as a viable strategy for managing CMD in cassava cultivation. However, the long-term effects of SA treatment and its efficacy across different varieties under varying environmental conditions require further investigation. In this study, the strategic application of SA to combat CMD is directly linked to the enhancement of SAR, as SA is widely recognized as an inducer of this defense mechanism [43].

SA functions as a mobile signaling molecule, transported to uninfected areas via apoplastic or simplistic pathways to activate systemic immunity [44,45,46]. Recently, Kim and Lim [47] reported that the pH gradient and deprotonation may lead to the apoplastic accumulation of SA before its transportation into the cytosol following pathogen infection. Additionally, SA functions in association with other phytohormones such as JA, ethylene (ET), and abscisic acid, and interacts with genes, proteins, and transcription factors (TFs), including PRs and phytoalexins, to trigger SAR and induced systemic resistance [48].

SAR provides sustained protection, and the continued reduction in viral load and disease severity suggests that SAR was sustained throughout the treatment period, which aided in the effective management of the viral infection over time. This finding supports the notion that treatment with SA can enhance plant immunity, which makes it a valuable tool in integrated pest management strategies [31].

The production of higher quantities of SA in plants induces the generation of various substances that inhibit pathogens, including ROS and enzymatic antioxidants, and reduces the pathogen count. Sofy et al. (2021) [36] observed that the production of ROS and the expression of enzymatic antioxidants related to disease resistance increased in AMV-infected eggplants after 14 days of treatment with SA.

### 3.2. Relative Gene Expression Levels of Resistance Genes Following SA Treatment

In this study, the expression levels of nine genes, namely, *PR1*, *PDF2*, *WRKY59*, *PR9e*, *PR7f5*, *Hsp90.9*, *Hsf8*, *SPS1*, and *SYP121* were analyzed due to their association with SAR, especially after treatment with SA. Gene expression was analyzed following the foliar application of SA at varying concentrations and time intervals and compared to that of the control plants. The expression levels of *Hsf8*, *SYP121*, *PR7f5*, *PDF2*, and *WRKY59* gradually downregulated after 1 week of treatment with 100 or 200 mg/mL SA, and were comparable to those of the control group (Figure 4). In contrast, the expression levels of *SRS1*, *Hsp90.9*, and *PR9e* were significantly upregulated at 1 week post-treatment, and increased continually for up to 2 weeks post-treatment. Notably, the upregulation in the expression levels of these genes was sustained beyond 2 weeks, and the gene expression patterns were consistent between the groups treated with 100 and 200 mg/mL SA. Interestingly, analysis of the gene expression dynamics of *PR1*, especially after treatment with 200 mg/mL SA, revealed that its expression was reduced markedly at first, from 0 to 1 week post-treatment, but increased after 1 week, as depicted in Figure 4. In this study, the collection of leaf samples following was performed concurrently during the measurement of viral particles described in the previous section.

Regression analysis of the expression levels of the nine aforementioned genes revealed that the gene expression patterns of *Hsf8*, *SYP121*, *PR7f5*, *PDF2*, and *PR1* in the plants treated with SA did not differ significantly from those of the control group, and were downregulated in CMD-infected cassava plants (Appendix A). However, the expression levels of *PDF2* and *PR1* were upregulated following treatment with SA, compared to those of the control plants. We additionally observed that the gene expression levels of *Hsp90.9*, *WRKY59*, *SRS1*, and *PR9e*, which are involved in plant immunity, tended to increase after treatment with SA. A concentration of 100 mg/mL was found to be most effective in inducing these genes (Appendix A).

Malichan et al. [49] demonstrated that the expression levels of *SRS1*, *Hsp90.9*, and *PR9e* in the KU50 cultivar of cassava, which is tolerant to SLCMV, were downregulated seven days after inoculation with SLCMV via whiteflies. Our findings revealed that treatment with SA further upregulated the expression of *SRS1*, *Hsp90.9*, and *PR9e* at one week post-treatment, indicating an enhanced response in cassava plants. Recently, Wei et al. [50] identified that the expression of *MeHsp90.9* and the plant immune response is enhanced by the interaction of *MeHsp90.9* with *MeSRS1* (SHI-related sequence 1) and *MeWRKY20*, which promotes the transcriptional activation of other genes via SA. Consistent with this finding, the results of RT-PCR demonstrated that *SRS1* and *Hsp90.9* were upregulated in a coordinated manner, suggesting that SA can enhance gene expression at increasing concentrations and fortify disease resistance.

The *PR1* gene serves as a marker for monitoring the activation of the SA and JA pathways in *Arabidopsis* during pest/pathogen interactions [51]. In this study, we observed that *PR1* was downregulated 1 week post-treatment with 100 or 200 mg/mL SA. However, the expression of *PR1* increased continually after 1 week of treatment with 200 mg/mL SA. Interestingly, Irigoyen et al. [52] observed that 10 PR genes, except *PR1*, of the COL2246 variety of cassava, were upregulated at 0, 0.5, 1, 2, 4, 8, 12, and 24 h post-treatment with SA. We propose that the genotype of cassava, concentration of SA, and time of sample collection influence the gene expression levels.

WRKY TFs play a crucial role in plant responses to various environmental stresses. In particular, the *WRKY59* gene is known to be induced following treatment with SA and plays a major role in fungal resistance. The *WRKY59* gene is present in several plant species and can be induced by SA [53], and has been shown to function as a resistance gene against *Ralstonia solanacearum* [54]. It has been additionally reported that the *WRKY* gene can suppress the expression of genes associated with disease severity in chili peppers [55]. Using yeast one-hybrid assays, a previous study identified that *PtrWRKY18* and *PtrWRKY35* are potential targets of *WRKY89* [56]. The *OsWRKY89* gene of rice has been shown to enhance resistance to rice blast fungus and white-backed plant hopper, and induce the production of SA [57]. In this study, we observed that the application of 100 or 200 mg/mL SA upregulated the expression of *WRKY59* from 0 to 1 weeks post-treatment, following which its expression was downregulated. The sustained upregulation of *WRKY59* for 1 week may account for the upregulation of the *PR1* gene after 1 week of treatment with SA (Figure 4). This finding suggested that *WRKY59* interacted with *PR1* to induce resistance against SLCMV in response to treatment with SA. This finding is consistent with the study by Wu et al. [58], which reported that the receptors of plant SA signaling, regulated by the *PR1* gene (NPR1), interact with the NPR3 and NPR4 proteins. Although NPR proteins interact with the SA signal, they cannot directly bind to DNA, and therefore require TFs to regulate the expression of downstream SA-responsive genes. Further studies are therefore necessary for exploring the interactions between *WRKY59* and *PR1* during SA IR to SLCMV.

### 3.3. Identification of Metabolites Associated with SAR

The results of metabolomic analysis revealed that L-serine was consistently detected across the plants treated with 0, 100, or 200 mg/mL SA at 0, 1, and 2 weeks post-treatment, and that there were no significant differences among the different treatment groups. Previous studies have demonstrated that L-serine serves as a primary substrate in plant tissues [59,60]. The results of KEGG analysis revealed that L-serine is a component of the glycine, serine, and threonine metabolism pathway; however, this pathway does not directly regulate SAR and likely contributes to myricetin synthesis.

The levels of L-threonine and myricetin were upregulated in cassava plants treated with 100 or 200 mg/mL SA at 1 and 2 weeks post-treatment; however, they were not detectable in untreated cassava plants. Statistical analyses revealed that there were no significant differences in the levels of L-threonine and myricetin among the different SA-treated groups (*p* < 0.05; Figure 5). The results of KEGG pathway analysis revealed that L-threonine was involved in glycine, serine, and threonine metabolism (Figure 6), while myricetin was associated with the flavonoid biosynthesis pathway.

Although limited, existing studies suggest that threonine may enhance resistance to pathogens, thereby potentially serving as a precursor for homoserine, which regulates the ET, JA, and SA pathways in response to downy mildew (*Hyaloperonospora arabidopsis*) in *Arabidopsis* sp. [61]. This implies that amino acids may trigger the defense response; however, the precise underlying mechanisms remain unclear. Our findings suggest that treatment with SA may also induce the defense response via amino acids. L-threonine also plays a crucial role in plant growth, development, stress tolerance, and gene regulation [62].

The results of KEGG pathway analysis indicated that myricetin is synthesized through the flavonoid biosynthesis pathway, which originates from kaempferol, a derivative of p-coumaroyl-CoA in the phenylpropanoid biosynthesis pathway (PPP) (Figure 7). L-phenylalanine was detected in all the SA-treated plants, and its levels increased 2 weeks post-treatment (Figure 8). Plant viruses activate SA biosynthesis and the production of other defense metabolites via PPP to initiate SAR. For instance, sugarcane mosaic virus triggers this response in maize [63]. Previous studies have identified various antiviral metabolites in TMV-infected tobacco and tomato plants [63,64]. In particular, it has been reported that 5-O-caffeoylquinic acid and quercetin accumulate at the site of TMV infection, and kaempferol was detected in distant tissues exhibiting SAR [65].

## 4. Conclusions

SA plays an important role in stimulating plant defenses against various pathogens via activating key signaling molecules in the plant immune system. SA induces a critical mechanism referred to as SAR, which enhances resistance to pathogens by inducing the expression of PRs. This response reduced the severity of CMD in infected cassava plants, as visually indicated by the absence of foliar symptoms at one and two weeks post-treatment. Additionally, SA treatment significantly reduced SLCMV loads in infected cassava plants. The alterations in gene expression following SA treatment suggest that SA induces the expression of immune-related genes, including *HSP90*, *WRKY59*, *SRS1*, and *PR9e,* in diseased cassava plants. In conclusion, 200 mg/mL SA was the most effective concentration for reducing viral particles in cassava plants, whereas 100 mg/mL SA was particularly effective in upregulating the expression of immune-related genes compared to control plants. Furthermore, SA enhances the regulation of secondary metabolite pathways involving L-serine within the glycine, serine, and threonine metabolism pathway, as well as the PPP. The findings obtained herein provide valuable insights for enhancing plant immunity and mitigating viral infections, especially in regions such as Thailand where there is a scarcity of healthy planting materials, and cassava farmers continue to utilize infected stems. The strategic application of SA at optimal concentrations could serve as a viable strategy for disease management and ensuring the sustainable cultivation of cassava.

## Figures and Tables

**Figure 1 metabolites-15-00261-f001:**
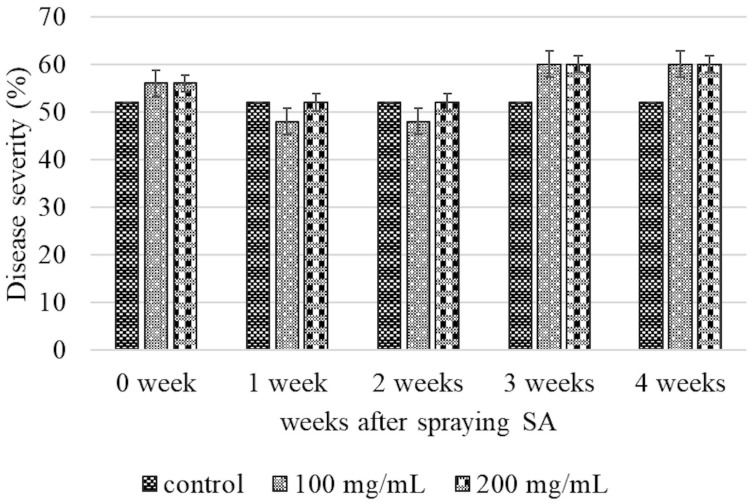
Determination of disease severity using SLCMVinfected cassava leaves following the foliar application of SA. Error bars indicated standard deviation, and columns not significant (*p* > 0.05) in the disease severity by one-way analysis.

**Figure 2 metabolites-15-00261-f002:**
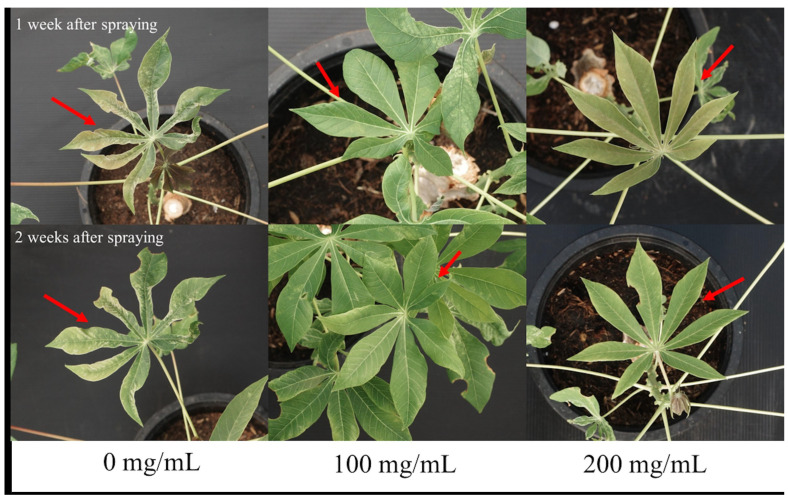
Phenotype of SLCMV−infected cassava plants following treatment with SA at 1 and 2 weeks post−treatment. Red arrows indicate symptoms on the same leaf in 0, 100 and 200 mg/mL SA.

**Figure 3 metabolites-15-00261-f003:**
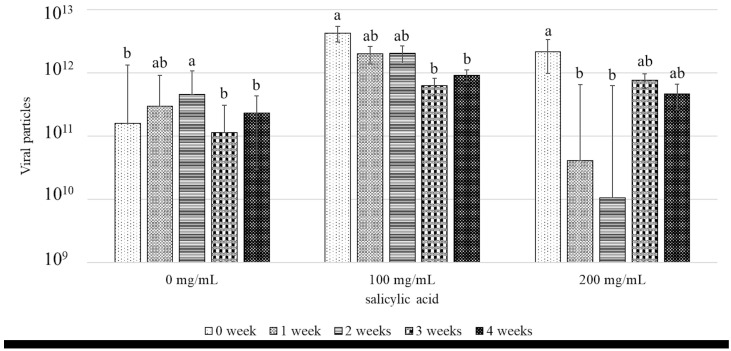
SLCMV particles in CMD−infected cassava plants following treatment with SA as determined by qPCR. Error bars indicated standard deviation, and columns with different letters indicate significant difference (*p* ≤ 0.05).

**Figure 4 metabolites-15-00261-f004:**
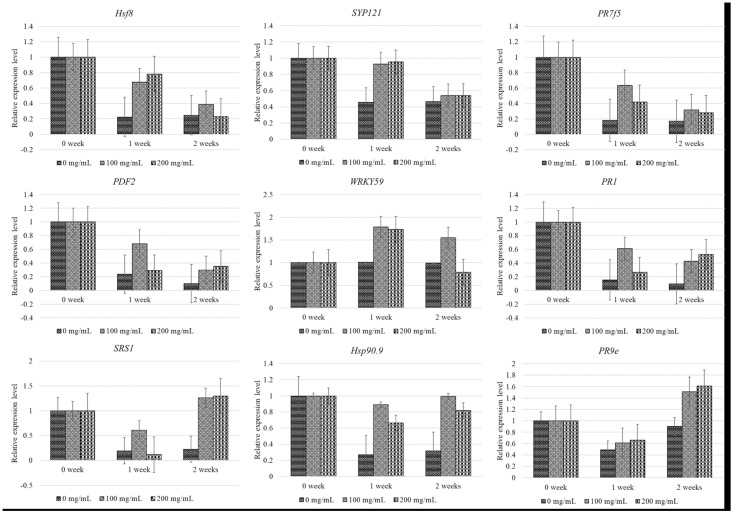
Expression patterns of nine genes related to the induction of plant immunity in SLCMV−infected cassava plants following the application of SA. Error bars indicated standard deviation.

**Figure 5 metabolites-15-00261-f005:**
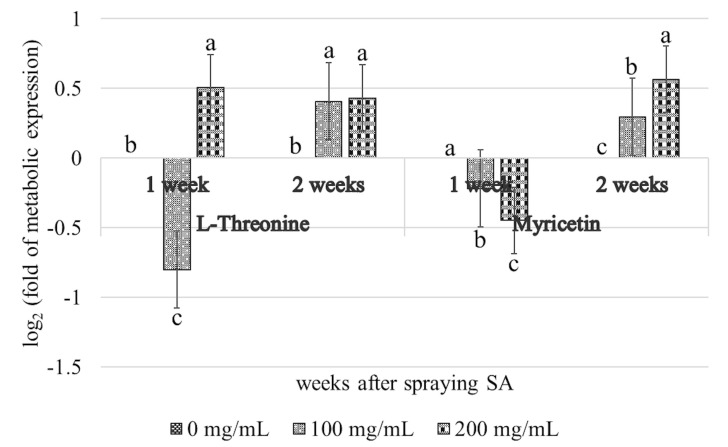
Expression patterns of L−threonine and myricetin metabolites in SLCMV−infected cassava plants following the application of SA. Error bars indicated standard deviation, and columns with different letters indicate significant difference (*p* ≤ 0.05).

**Figure 6 metabolites-15-00261-f006:**
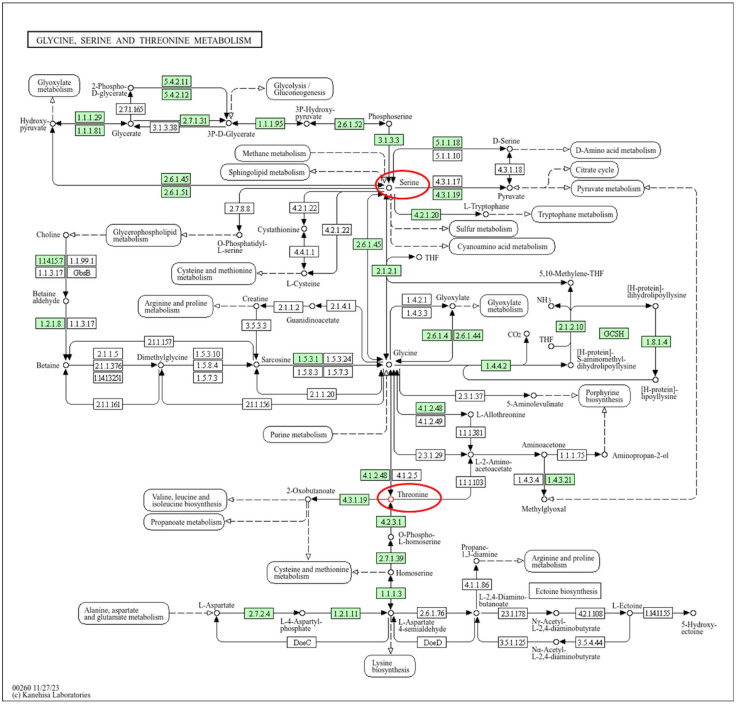
The glycine, serine, and threonine metabolism pathway illustrated using the KEGG Orthology within the KEGG pathway database. The red circles indicate the secondary compounds detected in the samples and the green boxes indicate to organism-specific pathway in KEGG pathway.

**Figure 7 metabolites-15-00261-f007:**
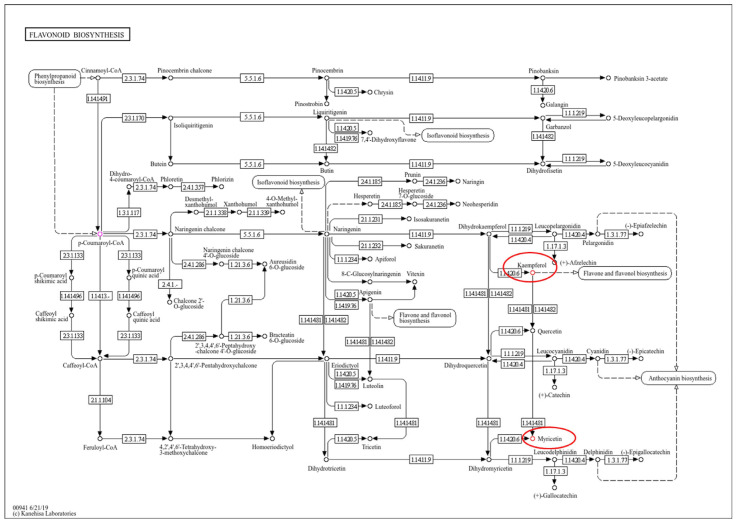
Flavonoid biosynthesis pathway illustrated using KEGG Orthology within the KEGG pathway database. The red circles indicate the secondary compounds detected in the samples.

**Figure 8 metabolites-15-00261-f008:**
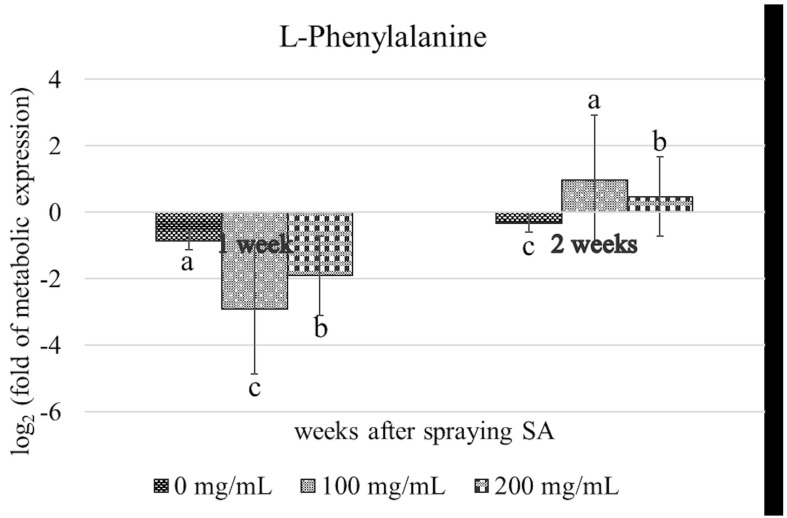
Expression patterns of L−phenylalanine metabolites in SLCMV−infected cassava plants following the application of SA. Error bars indicated standard deviation, and columns with different letters indicate significant difference (*p* ≤ 0.05).

## Data Availability

The data that support the findings of this study are available from the corresponding author upon reasonable request.

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
