# Peer review of "Enhancing Plant Resistance to Sri Lankan Cassava Mosaic Virus Using Salicylic Acid"

_metabolites, 2025, doi:10.3390/metabo15040261_

Round 1
Reviewer 1 Report
Comments and Suggestions for Authors
In Material and Method
1. How can plants be infected with CMD? Give detailed information.
2. For SA application, how many times did SA apply to leave? For example, three times a week?
3. Which statistical program was used for disease severity?
4. Was universal primer used for quantification of SLCMV particles? It must be written accession number for AV1 gene and a reference for primer sequence. If primers were designed by the author, which program was used for designing?
5. For each analysis, please add statistical information, e.g., disease severity, metabolomic, etc.
In Results and Discussion
1. Add detailed info in the Figure titles, e.g. what is the meaning of red arrow, what are the meanings of a, b, ab? And add statistical results.
Author Response
Reviewer 1
In Material and Method
- How can plants be infected with CMD? Give detailed information.
ANSWER
Thank you for your question. Cassava mosaic disease (CMD), caused by Sri Lankan cassava mosaic virus (SLCMV), can infect plants through two primary routes: (1) the use of infected planting materials (such as stem cuttings), and (2) transmission by insect vectors, particularly the tobacco whitefly (Bemisia tabaci). In this study, CMD infection was established using infected planting materials to ensure consistent and reliable infection across all experimental replicates. Additional details regarding the infection method have been provided in Lines 58–67 of the revised manuscript.
- For SA application, how many times did SA apply to leave? For example, three times a week?
ANSWER
Thank you for your question regarding the frequency of SA application. In this experiment, SA was applied as a foliar spray only once. This decision was based on previous findings indicating that excessive application of SA may lead to phytotoxic effects in plants. Therefore, a single application was chosen to minimize potential negative impacts while still allowing us to evaluate its effectiveness in inducing systemic acquired resistance (SAR).
- Which statistical program was used for disease severity?
ANSWER
Thank you for your question. In this experiment, disease severity data were analyzed using one-way analysis of variance (ANOVA) performed in RStudio. Significant differences between treatment means were determined using the Least Significant Difference (LSD) test, with a significance threshold set at p ≤ 0.05.
- Was universal primer used for quantification of SLCMV particles? It must be written accession number for AV1gene and a reference for primer sequence. If primers were designed by the author, which program was used for designing?
ANSWER
Thank you for your comment regarding the quantification of SLCMV. The primers used for detection and quantification of the SLCMV AV1 gene were specifically designed for this study using the Primer3 software. The reference sequence used as a template for primer design was obtained from the NCBI GenBank database (accession number: MN577580, MN577579, MN544647 and MN577578). These primers were not universal but were custom-designed to specifically amplify a region of the AV1 gene.
- For each analysis, please add statistical information, e.g., disease severity, metabolomic, etc.
ANSWER
Thank you for your valuable recommendation. We have now added detailed statistical information for each analysis, including disease severity and metabolomic data, in the revised manuscript (Lines 280–296).
In Results and Discussion
- Add detailed info in the Figure titles, e.g. what is the meaning of red arrow, what are the meanings of a, b, ab? And add statistical results
ANSWER
Thank you for your valuable recommendation. We have added detailed descriptions in the figure titles, including the meaning of red arrows and the notations "a", "b", "ab", which indicate statistically significant differences. Additionally, statistical results have been included where appropriate in all relevant figure legends.

Reviewer 2 Report
Comments and Suggestions for Authors
please read the file

Author Response
Reviewer 2
Increasing plant resistance to viruses using salicylic acid is common worldwide. The mechanisms are well studied. For cassava plants, SA-induced immunity to the virus is studied for the first time. The work studies the genes and pathways that are involved in phytohormone during viral infection. This is correct.
In the abstract, add the importance of a crop like cassava to the country. The degree of infection with these viruses to highlight the relevance and necessity of the study
- Please explain in the Introduction why you took such concentrations of phytohormone? Is there any literature on the use of such concentrations of salicylic acid? Can it be sprayed?
ANSWER
Thank you for your comment. The SA concentrations (0, 100, and 200 mg/mL) were chosen based on previous studies. Kidulile et al. (2018) found that higher SA concentrations (0-40 mg/L) used in MS medium led to more regenerated plants but lower survival rates. Similarly, Kung'u et al. (2024) observed reduced plant growth when cassava stems were soaked in SA at 1.25-5 mM, indicating that high concentrations can be toxic to plants. In contrast, Zhang and Li (2019) and Xi et al. (2021) reported that SA promoted plant growth, suggesting that the effect of SA might depend on host-virus interactions.
Although this is the first study to apply SA to CMD-infected cassava, previous research shows that SA sprays can enhance disease resistance in crops like tomato, tobacco, and eggplant (Basit et al., 2021; Sofy et al., 2021; Wang et al., 2022). A preliminary test showed that 50 mg/mL of SA did not alter virus levels, but concentrations of 500 mg/mL and 100 mg/mL caused toxicity, including yellowing and leaf drop, consistent with Kidulile et al. (2018).
- What stage of plant development?
ANSWER
Thank you for your question. In this experiment, we used cassava plants that were 6 weeks old. By week 6, cassava plants are generally in the early stages of vegetative growth. At this stage, the plants are focusing on stem elongation, leaf development, and establishing a strong root system, which will be critical for future tuber production.
- Can cassava tubers be treated with salicylic acid before planting? Please pay attention to how cassava multiplies. Does this create difficulties in reducing the viral load?
ANSWER
Thank you for your recommendation and concern. We did a preliminary experiment where cassava stems were soaked in salicylic acid before planting. However, the survival rate of the plants was very low, indicating that pre-planting treatment with salicylic acid might not be effective or could cause phytotoxicity, which makes it challenging to reduce the viral load. Therefore, we opted to apply salicylic acid post-planting in this experiment.
- Convert the concentration of salicylic acid to micromoles
ANSWER
Thank you for your concern. The concentrations of salicylic acid were converted as follows:
- 100 mg/L of SA is equivalent to 0.72 mM
- 200 mg/L of SA is equivalent to 1.44 mM
The results and their discussion are described in part in accordance with the results obtained. The discussion is complete. All the data provided prove the effectiveness of using SA to increase resistance to the virus.
I am concerned about how much SA treatment affects the endogenous SA content. Is it possible to discuss the literature data? Do they exist at all? Please include this. Since the metabolites of secondary metabolism are considered, the phytohormone itself should also be discussed.
ANSWER
Thank you for your valuable comment. You raise an important point regarding the potential effects of exogenous SA treatment on the endogenous SA content in plants. Indeed, several studies have shown that the application of exogenous SA can influence the plant’s endogenous SA levels, although the extent of this effect may depend on factors such as plant species, concentration of SA applied, and the duration of exposure.
For instance, some studies have reported that applying exogenous SA can lead to an increase in endogenous SA, thereby enhancing the plant’s defense mechanisms. On the other hand, higher concentrations of SA can sometimes lead to a feedback mechanism that reduces endogenous SA production. However, it is also suggested that the impact of exogenous SA might vary based on the plant’s developmental stage and its inherent capacity to regulate hormonal levels.
Unfortunately, literature specific to the effect of exogenous SA treatment on endogenous SA content in cassava is limited, but there are similar reports in other crops like tomato and tobacco (Basit et al., 2021). Further investigations are necessary to explore the dynamics of SA regulation in cassava during virus-induced stress and hormonal treatment. (Line no. 317-325)
